# Metataxonomics and Metabolomics Profiles in Metabolic Dysfunction-Associated Fatty Liver Disease Patients on a “Navelina” Orange-Enriched Diet

**DOI:** 10.3390/nu16203543

**Published:** 2024-10-18

**Authors:** Francesco Maria Calabrese, Emanuela Aloisio Caruso, Valentina De Nunzio, Giuseppe Celano, Giuliano Pinto, Miriam Cofano, Stefano Sallustio, Ilaria Iacobellis, Carmen Aurora Apa, Monica Santamaria, Maria Calasso, Gianluigi Giannelli, Maria De Angelis, Maria Notarnicola

**Affiliations:** 1Department of Soil, Plant and Food Science, University of Bari Aldo Moro, 70126 Bari, Italy; giuseppe.celano@uniba.it (G.C.); stefano.sallustio@uniba.it (S.S.); ilaria.iacobellis@uniba.it (I.I.); carmen.apa@uniba.it (C.A.A.); monica.santamaria@uniba.it (M.S.); maria.calasso@uniba.it (M.C.); maria.deangelis@uniba.it (M.D.A.); 2Laboratory of Nutritional Biochemistry, National Institute of Gastroenterology IRCCS “Saverio de Bellis”, 70013 Castellana Grotte, Italy; emanuela.caruso@irccsdebellis.it (E.A.C.); valentina.denunzio@irccsdebellis.it (V.D.N.); giuliano.pinto@irccsdebellis.it (G.P.); miriam.cofano@irccsdebellis.it (M.C.); 3Scientific Direction, National Institute of Gastroenterology IRCCS “Saverio de Bellis”, 70013 Castellana Grotte, Italy; gianluigi.giannelli@irccsdebellis.it

**Keywords:** metabolic dysfunction-associated fatty liver disease, orange diet, fecal metabolomics, 16S metataxonomics

## Abstract

Background/Objectives: Metabolic dysfunction-associated fatty liver disease (MAFLD) is currently the most common cause of chronic liver disease. Systemic inflammatory status and peripheral metabolic symptoms in the clinical picture have an impact on gut commensal bacteria. Methods: Our designed clinical trial was based on a cohort of patients with MAFLD whose diet included the daily consumption of 400 g of “Navelina” oranges for 28 days, compared with a control group of patients with the same pathologic conditions whose diet did not include the consumption of oranges and other foods containing similar nutrients/micronutrients. We used 16S metataxonomics and GC/MS analyses to identify taxa and urine/fecal VOCs, respectively. Results: A set of micronutrients from the diet were inspected, and some specific fatty acids were identified as the main contributors in terms of cluster sample separation. Metataxonomics and metabolomics profiles were obtained, and a stringent statistical approach allowed for the identification of significant taxa/VOCs, which emerged from pairwise group comparisons in both fecal and urine samples. Conclusions: In conclusion, a set of taxa/VOCs can be directly referred to as a marker of dysbiosis status and other comorbidities that, together, make up the pathologic burden associated with MAFLD. The investigated variables can be a target of therapeutic strategies.

## 1. Introduction

Non-alcoholic fatty liver disease (NAFLD), now known as metabolic dysfunction-associated fatty liver disease (MAFLD), is the most common cause of chronic liver disease worldwide, with a prevalence of 20–30% in Western countries [1]. In current clinical practice, MAFLD is diagnosed based on at least one of three pathological conditions: obesity/overweight (BMI ≥ 25 kg/m^2^), type 2 diabetes mellitus (T2DM), and/or metabolic dysregulation. Therefore, MAFLD, unlike the term NAFLD, emphasizes metabolic risk and focuses on alterations in glucose (insulin resistance) and lipid metabolism (lipotoxicity, oxidative stress, etc.), as well as the significant role of inflammatory processes in hepatocytes [2,3].

It is known from the literature that the consumption of polyphenol-rich foods, such as oranges, may positively modulate both the microbiota and NAFLD and other metabolic and non-metabolic diseases, including obesity, dyslipidemia, insulin resistance, and type 2 diabetes, as well as cancer and atherosclerosis. This is due to their hypolipidemic, antioxidant, and anti-inflammatory properties. Thus, they are currently considered promising nutraceutical agents for the handling of these pathological conditions [4,5,6].

Flavonoids are bioactive compounds which are widely present in a variety of foods, including fruits and vegetables, and beverages like tea, coffee, and wine [4]. In particular, the flavonoids found in citrus fruits are most concentrated in the albedo and the membranes dividing the segments; these are hesperidin, naringin, naringenin, nobiletin, and tangeretin, among which hesperidin is the most abundant [7,8,9]. As stated in both in vitro and in vivo studies, recent evidence suggests that the flavonoid hesperidin may potentially improve NAFLD by exerting hypoglycemic effects, promoting fatty acid β-oxidation through the activation of SIRT1/PGC1α, and ultimately impacting lipid profiles [10]. Similarly, the treatment of patients with metabolic syndrome with this molecule for 3 weeks was shown to significantly reduce circulating concentrations of high-sensitivity C-reactive protein, serum amyloid protein A, and soluble E-selectin [11]. In another clinical study, orange juice intake was shown to have a positive impact on plasma lipid profiles, with reduced levels of total triglycerides and total cholesterol and improved LDL-C and VLDL concentrations in subjects with obesity and insulin-resistance [12]. However, despite media concerns, daily consumption of orange juice does not cause an increase in dietary sugar intake and body weight [13,14]. Also, in vitro, flavonoids extracted from the “Tacle” orange variety show an inhibitory action on cholesterol synthesis [15].

In general, under normal conditions, adipose tissue stores lipids in the form of triglycerides, whereas during obesity, hyperlipidemia causes excessive macrophage infiltration in adipose tissue and the liver, resulting in the production of proinflammatory cytokines such as TNF-α, IL-6, and iNOS [16,17], which are in turn associated with systemic inflammation and atherogenesis [18,19]. Animal studies demonstrated a protective effect of flavonoids against the inflammation associated with obesity and atherosclerosis. Diabetic rats or rats with liver fibrosis treated with naringin or hesperidin showed a reduction in inflammatory cytokines such as TNFα, IL-6, CRP, and IL-1β and an increase in anti-inflammatory cytokines such as IL-10 and IL-13 [20,21].

The aim of this study is to evaluate the effects and changes at the metabolomics and metagenomics levels following the daily intake of “Navelina” oranges in subjects at high metabolic risk, such as patients with MAFLD.

We here report the results of fecal metataxonomics and metabolomics analyses of both fecal and urine samples conducted in the framework of a two-arm randomized clinical trial, where the first arm was composed of patients with MAFLD following a dietary regimen supplemented with fresh oranges and the second included control patients whose balanced diet did not include oranges or foods containing the same excipients/chemical compounds as oranges. The stringent statistical analyses here conducted sum up important variables derived from the diet plus specific VOC markers from urine and fecal samples.

## 2. Materials and Methods

### 2.1. Patient Recruitment

Sixty-two subjects with MAFLD were recruited on a voluntary basis by the Ambulatory of Nutrition of our Institute (National Institute of Gastroenterology “S. de Bellis”) from February 2023 to November 2023. Trial inclusion criteria were as follows: age ranging from 30 to 65 years, confirmed hepatic steatosis (Controlled Attenuation Parameter (CAP) score > 270 dB/m), and one of the following metabolic criteria: overweight/obesity, type 2 diabetes, and/or metabolic dysregulation. Exclusion criteria included gastroesophageal reflux, chronic inflammatory diseases, oncological diseases, serious medical conditions, special diets, taking anticoagulants, and religious reasons.

All subjects were told about the purpose, procedures, and risks of this study. They were instructed not to change their lifestyle and they were not given a weight loss goal. Written informed consent was obtained from all subjects for blood sample and clinical data collection. This study was designed in accordance with the Helsinki Declaration. This clinical study was registered at ClinicalTrials.gov (http://www.clinicaltrials.gov, accessed on 22 August 2024) (NCT05558592).

### 2.2. Study Design

At baseline (T0), patients underwent a nutritional examination. Nutritionists explained how to fill in a daily diary of the patients’ eating habits using a photo atlas and a physical activity questionnaire (IPAQ), to be handed in at the next visit (T1–7 days). All of the patients underwent anthropometric measurements, and a venous blood sample was delivered together with one stool and one urine sample. After 7 days (baseline visit, T1), patients were randomly assigned to two study groups. More specifically, a computer-generated sequence of non-unique and unsorted numbers within the range from 1 to 2 was used to physically allocate samples in the control group and the treated group. The orange group consisted of 31 subjects (24 males and 7 females) who were invited to take a daily consumption of 400 g of “Navelina” oranges, calculated as net of waste/die/per person, for 28 days. Each subject received a quantity of oranges sufficient to cover the entire treatment (about 12 kg). The 31 subjects (21 male and 10 female) in the control group had to follow an enriched diet with a daily consumption of 400 g of fruits, with the exception of citrus fruits and foods rich in vitamin C, for 28 days. All study subjects received dietary recommendations, including limited alcohol, caffeine, and foods rich in vitamin C (Appendix A). To ensure that no major changes in lifestyle occurred, all participants were asked to complete a food diary and an IPAQ questionnaire throughout the entire study. At the end of the clinical trial (final visit, T2), all subjects returned to the Ambulatory of Nutrition delivering the above-mentioned requested materials and on the same day underwent the nutritional visits. Research dietitians reviewed diaries side by side with participants, and soon after they analyzed the collected material using the MetaDIETA Professional software version 4.0.1 (Meteda, Rome, Italy).

Fecal and urine samples were gathered at two times: before and at the end of the clinical trial.

### 2.3. DNA Fecal Extraction and Pooled Library Preparation

A total of 15 control samples and 19 orange-treated samples were chosen. After the extraction of DNA, and before the creation of the pooled library, one sample from the treated group was excluded because it did not reach the minimum DNA concentration required. An aliquot (500 µL) of each sample was diluted in 1 mL of PBS-EDTA (phosphate buffer 0.01 M, pH 7.2, 0.01 M EDTA) and centrifuged (14,000× *g* at 4 °C for 5 min). The obtained pellet was washed twice. The next steps of the extraction were performed following the protocol of FastDNA^®^ Pro Soil-Direct Kit (MP Biomedicals, Thomas Irvine, CA, USA). The quality check and DNA concentration were measured using a NanoDrop^®^ ND-1000 Spectrophotometer (ThermoFisher Scientific Inc., Milan, Italy) at 260, 280, and 230 nm, and using a Qubit 3 fluorometer (ThermoFisher Scientific Inc., Milan, Italy). The extracted DNA was standardized at 100 ng/µL and pooled before the PCR steps.

### 2.4. DNA Amplification and Sequencing

V3-V4 bacterial 16S ribosomal RNA (rRNA) gene was amplified and libraries were sequenced on a Miseq2 Illumina platform, part of the facilities available at Di.S.S.P.A department (University of Bari). The first PCR was performed by preparing 25 μL mixes using 12.5 μL of Kapa HiFi HotStart ReadyMix Taq 2× (Kapa Biosystems, Wilmington, MA, USA), 5 μL for each primer (1 μM), 2.5 μL of sample DNA. Twenty-five cycles of 30 s denaturation (95 °C), 30 s primer annealing (55 °C), and 30 s primer elongation (72 °C) were performed, followed by a final elongation step (72 °C) of 5 mins. A 1.5% agarose gel was prepared to verify DNA amplification.

PCR amplicons were purified using an Agencourt AMPure kit (Beckman Coulter, Milan, Italy), and the resulting products were labeled using the Nextera XT index kit (Illumina Inc., San Diego, CA, USA), according to the manufacturer’s instructions.

After quality control and quantification, the indexed libraries were normalized, pooled to the required molarity, and then loaded into the sequencing platform together with buffers, reagents, and an internal PhiX control (PhiX Control Kit v3). Before pooling, single libraries were normalized to a concentration of 4 ng/µL. A volume of PhiX equal to 15% of the final pooled library was added.

### 2.5. Bioinformatics Analyses, Filtering, and Annotation

PCR primers and Illumina adapters were removed by means of an ad hoc developed script enclosing the Cutadapt tool [22]. The sequence’s quality was assessed using the demux plugin and the FastQC Version 0.12.0 (http://www.bioinformatics.babraham.ac.uk/projects/fastqc/, accessed on 3 September 2024) software. Accordingly, sequences were denoised and ASVs were obtained through the Qiime2 deblur plugin.

Suitable QIIME2 plugins were used also for the following steps. A V3-V4 specific classifier, formatted for use in Qiime2, was built starting from SILVA release 138. This database was filtered for the following reasons: (i) to remove low-quality data, (ii) to select sequences based on minimum length and taxonomy, and (iii) to dereplicate them. Only the V3-V4 regions were then extracted using an in silico PCR using the following primer couple: 341F Illumina 5′CCTACGGGNGGCWGCAG3′ V3-V4 forward, and 805R Illumina 5′GACTACHVGGGTATCTAATCC3′ V3-V4 reverse. The obtained V3-V4 collection was trained to obtain the QIIME2 compliant classifier that was used to taxonomically identify the ASVs. Unassigned, mitochondrion, and chloroplast ASVs were removed. A taxa relative abundances table and rarefaction curves were obtained by means of ad hoc Python and R scripts, respectively. Several core diversity metrics, including alpha and beta diversity, were obtained.

### 2.6. Statistical Analyses

Anthropometric and clinical results were presented as means plus standard deviations (M ± SD) for continuous variables at T1 and T2. The Wilcoxon matched-pairs signed-rank test was applied for continuous parameters to evaluate changes between T1 and T2 in the control and treated groups. Statistical test significance was set at *p* < 0.05. All analyses on clinical data were performed using StataCorp 2023 software, version 18 (College Station, TX, USA: StataCorp LLC).

The complete matrices of genus and VOC abundances were inspected by means of a discriminant analysis of principal component (DAPC) using the R “adegenet” package v2.1.120 trying to cluster samples based both on an “a priori” and “a posterior” group belonging. The a priori hypothesis was inspected without superimposing any metadata grouping condition and using the ‘find.clusters’ clustering algorithm. Metabolic pathway predictions were obtained from 16S rRNA abundance matrix using Picrust version 2.0 software, run as a plugin within the QIIME2 library.

We performed an exploratory factor analysis using the principal component factor model. This allowed us to reduce the number of variables by describing the linear combinations of the variables that contain most of the information and that admit meaningful interpretations. To restrict the entire panel to factors with a strong clinical interpretation, we a priori picked eigenvalues equal to or greater than two. A graphical assessment of the eigenvalues was performed. We used post-estimation tools such as orthogonal rotation (varimax) to improve statistical weight imputed to large initial loadings. Finally, the Kaiser–Meyer–Olkin (KMO) measure of sampling adequacy was applied.

To retrieve significant changes in taxa, a BH-corrected Welch test joined with a fold change analysis was run between the thesis groups. Statistically significant variables in the useful pairwise group comparisons were graphically rendered as volcano plots.

### 2.7. Fecal and Urinary Metabolomics

Gas chromatography/mass spectrometry profiles from untargeted volatile organic compounds from both urine and fecal specimens were run on 20 coupled samples (T1 and T2) from orange treated patient set plus 16 coupled samples from the control group.

For fecal metabolomics analyses, we started from an aliquot (1 g) of each fecal sample collected in the in vivo study. It has been fortified with 10 μL of internal standard solution (4-methyl-2-pentanol) with a concentration of 33 ppm and placed in 20 mL glass vials sealed with silicon rubber septa with polytetrafluoroethylene (PTFE) coating (20 mm diameter) (Supelco, Bellefonte, PA, USA) [23].

GC-MS (gas chromatography-mass spectrometry) analysis was performed using a Clarus 680 (Perkin Elmer, Beaconsfield, UK) equipped with a Rtx-WAX capillary column (30 m × 0.25 mm i.d., 0.25 μm film thickness) (Restek, Bellfonte, PA, USA). The column temperature was initially set at 35 °C for 8 min with a thermal gradient of 4 °C/min up to a temperature of 60 °C, 6 °C/min up to 160 °C and finally 20 °C/min up to 200 °C, with the latter temperature maintained for 15 min. Helium was used as carrier gas, maintaining a continuous flow of 1 mL/min. The gas chromatography system was coupled with a Clarus SQ 8C single quadrupole mass spectrometer (Perkien Elmer). The source and interface temperatures were maintained at 250 °C and 230 °C, respectively.

A 20 mL glass vial containing 2 g of of acidified (pH 2) urine sample with 1 g of NaCl and 10 μL of internal standard (4-methyl-2-pentanol) was sealed. To obtain the best extraction efficiency, the solid-phase microextraction (SPME) was performed by exposing a conditioned 75 μm Carboxen/PDMS fiber (Supelco, Bellefonte, PA, USA). The extracted compounds were desorbed in splitless mode for 3 min at 280 °C. The analysis was carried out using a Clarus 680 gas chromatograph (PerkinElmer, Waltham, MA, USA) equipped with an Elite-624Sil MS Capillary Column (30 m × 0.25 mm i.d.; 1.4 μm film thickness). The column temperature was set initially at 40 °C for 3 min and then increased to 250 °C at 5 °C/min and to 280 °C at 10 °C/min and finally held for 5 min.

Electron ionization masses were recorded at 70 eV in the mass-to-charge ratio interval, which was *m*/*z* 34 to 350 in total ion chromatogram (TIC) scanning mode. The gas chromatography-mass spectrometry (GC-MS) analysis produced a chromatogram, with peaks corresponding to individual compounds. Compound identification was performed by matching the spectra to the National Institute of Standards and Technology (NIST) 2008 library, with a match score threshold of >85% and a peak area greater than 1,000,000. Additionally, the Automated Mass Spectral Deconvolution & Identification System (AMDIS) was used as an effective tool for analyte identification.

4-Methyl-2-pentanol (1 mg/L final concentration) was used as an internal standard to quantify identified compounds by comparing the relative areas to the internal standard area.

## 3. Results

Two different dietary regimens were compared in patients suffering from MAFLD: in the treated group patients received a dietary supplementation with fresh oranges, whereas the control group received no citrus fruit. Both the groups were asked to avoid the consumption of other foods enriched in vitamin C (Appendix A). The set of anthropometric variables, plus the micronutrient dietary intake, was investigated by means of DAPC and rotating factor analyses. Both the metataxonomic and metagenomic profiles were used to elucidate those taxa and volatile organic compounds (VOCs) associated with MAFLD samples under the orange-supplemented diet or the control one.

We first ran a discriminant analysis of principal components on anthropometric/micronutrient intake. The a priori DAPC showed how the entire set of samples can be potentially ascribed to four different groups, as explained by the Bayesian information criterion (BIC) curve (Figure 1A,B). When the a posterior group assignment was used, samples after the orange intervention were placed apart from all of the other groups that almost completely overlapped in the second and third quadrants (Figure 1C). Looking at the DAPC loading plot (Figure 1D), stearic, arachidonic, and palmitic acids, together with SFA, are the variables that much more impacted the DAPC cluster positions. We compared the “a priori” versus the “a posterior” group assignments, as graphically rendered in the DAPC assign plot (Figure 1E), and the orange-treated group appeared to be the most homogeneous, with only one sample exhibiting an uncertain assignation.

Variable weight was also evaluated based on a factor analysis by means of the “factanal” function in the built-in stats R package. The function performs a maximum-likelihood factor analysis on our anthropometric and nutrients/minerals data matrix (Appendix A). Three factors useful in grouping impacting variables were identified. Factors were evaluated on the basis of loading and uniqueness (Table 1). Thus, the here-implemented Bartlett’s weighted least-squares scores method identifies monosaturated and polyunsaturated AAs as part of the first factor as the ones with the highest loadings. These variables, at the same time, showed low uniqueness values (where 1 indicates commonality). Total fiber, calcium, phosphorous, thiamine, riboflavin, and niacin are part of the second factor, whereas lipids and carbohydrates are part of the third one. Noticeably, DAPC and rotating factor analyses agreed on recognizing a high statistical weight to specific variables; SFA, stearic, palmitic, and arachidonic acids emerging from DAPC are all part of the first factor.

### 3.1. Metataxonomics Diversity Metric Estimate

Looking at 16S metataxonomics annotation results, a total of ten phyla, eighteen classes, forty-one orders, ninety-one families, one-hundred-and-ninety-five genera, and three-hundred-and-sixteen species were annotated from filtered paired-end read sets (Appendix A—Sheet “Species”).

When the whole 16S genus taxa abundance matrix was used, the investigated total batch of samples did not reveal a significant change in terms of alpha and beta diversity richness when samples from the control and orange-supplemented subsets were compared at both sampling times (Appendix A).

### 3.2. Clustering Analyses on 16S Taxa at the Genus Levels

The PCA, DAPC, and PLS-DA analyses on 16S sequences at the species taxonomic level have too much background noise and did not result in a cluster separation based on a biological rationale.

Also, at first inspection, the partial least-squares discriminant analysis (PLS-DA), run on the total panel of genera, did not discriminate between the treated and control samples. Thus, because of the overlap among the four sample clouds (Appendix A), and in line with the micronutrient analyzed data, the clustering analysis was also run by using a second discriminant analysis, i.e., the DAPC algorithm, which maximizes the intergroup variance and which uses a Kmeans approach to find out the number of clusters in an unsupervised manner without superimposing sample group belonging.

As a matter of fact, the discriminant analysis of principal components allowed us to stratify samples (Figure 2A) based on the genus abundance matrix. Treated T2 samples placed apart from T1 and from patients who followed the control diet.

The DAPC loading plots are indicative of the most-contributing variables in terms of taxa at the genus level. The genera *Sarcina*, *Streptococcus*, *Flavonifractor*, *Lachnospiraceae UCG 003*, and *UCG 008*, *Eubacterium nodatum group*, *Clostridium methylpentosum group*, and *Rombustia* are in order of statistical weight in the genera over the 0.02 loading threshold.

### 3.3. Pairwise Group Comparison Genus Annotation

When the treated and control samples were compared after the dietary intervention at T2, the first group revealed an increase in *Metahonospera*, *Eggerthella*, and *Oscillospira*, whereas *Senegalimassilia*, *Negativibacillus*, and *Eubacterium ventriosum group* decreased (Figure 3). Three of these above-cited genera, i.e., *Senegalimassilia*, *Metahonospera*, and *Eubacterium ventriosum group*, also emerged in the case of the treated T2 group and were compared with the control samples.

### 3.4. Pathway Prediction from 16S Matrix

Based on 16S taxa presence/absence and abundances, we predicted the presence of biochemical pathways. The subsequent statistics of the groups did not yield any results after a multiple test correction.

### 3.5. Fecal and Urinary Metabolomics VOC Profiles

We both carried out metabolomic VOC profiles from fecal and urine samples. In this case, DAPC cluster stratification took aid from merging together the two matrices of variables. As shown in Figure 4, a DAPC plot similar to the one obtained when anthropometric and micronutrients was used. The set of urine and fecal VOCs that mostly impacted to sample cluster separation is reported as the loading plot and included those variables that impacted sample clustering, colored in red (urine) and blue (feces) based on specimen belonging.

We analyzed statistically significant VOCs from both fecal and urine metabolomics by fixing the time and comparing the two different study arms (T2 treated vs. T2 control samples), but also comparing samples in the treatment arm before and after orange consumption.

A total of three VOCs from urine were maintained after multiple test correction (FDR), thus distinguishing the two groups of samples in the treated arm (T2 vs. T1). More in detail, Camphene and 2-Propanol, 2-methyl- decreased after treatment, whereas Phenol, 2-methoxy-4-(1-propenyl)- increased. As reported in Table 2, when treated samples were compared against controls (T2), only a decrease in dodecane concentration was detected. As far as it concerns fecal metabolomics, the three VOCs resulted in a statistical difference in terms of concentration between treated samples and controls at T2. Precisely two of them increased after treatment (5-Hepten-2-one; 6-methyl-, 2,6-Octadienal, 3,7-dimethyl-, (E)-), whereas only one (1-Hexadecanol) decreased. The comparison between fecal samples (T2 treated versus both T1 treated and T2 controls) resulted in a list of four increased and twelve decreased VOCs, respectively. More precisely 5-Hepten-2-one, 6-methyl-; 2,6-Octadienal, 3,7-dimethyl-, (E)-; 5-Hepten-2-one, 6-methyl-; and Linalool increased at T2, whereas 1-Hexadecanol; Ethanone, 1-(1-cyclohexen-1-yl)-; 1H-Indole, 2-methyl-; Ketene; 2-Propanamine, 2-methyl-; Methyl tetradecanoate; Acetamide; Phenol, 2,6-dimethyl-; Dodecanoic acid, ethyl ester; Mequinol; Pyrazine, 2,6-dimethyl-; and Propanal, 2-methyl- decreased.

### 3.6. Statistical Correlation of Clinical, Metataxonomics, and Metabolomics Variables

Statistically significant genera and VOCs, resulting from the fold change analysis, based on the meta-taxonomics and untargeted metabolomics’ GC-MS data, respectively, together with the most-impacting clinical variables emerging from the rotating factor analysis, were inspected in terms of Pearson’s correlation. Appendix A reports all of the statistically significant variables derived from inter-group correlations, i.e., variables belonging to these following classes: (i) clinical measurements, (ii) metataxonomic assignment, and (iii) VOCs that resulted in statistical significance in the above-mentioned analyses. Few significative correlations emerged, and only some of these have a correlation value greater than 0.6. More in detail, *Negativibacillus* positively correlated with the mequinol metabolite and *Senegalimassilia* positively correlated with dodecanoic acid, ethyl ester, and ketene.

## 4. Discussion

The enrolled MAFLD patients followed a clinical trial based on a double-arm randomized experimental design in which ‘Navelina’ oranges were consumed daily. Subjects diagnosed with obesity/overweight (BMI ≥ 25 kg/m^2^), suffering from type 2 diabetes mellitus (T2DM) and/or metabolic dysregulation, were considered to be at high metabolic risk, and their hepatic steatosis was confirmed through a Controlled Attenuation Parameter (CAP) score higher than 270 dB/m.

Our workflow relapsed on a multiomics approach accounting for metataxonomics and metabolomics analyses both on urine and feces. According to the multiple hit theory, the microbiota and its relationship with diet are key factors that undoubtedly influence the development of NAFLD and its associated comorbidities [23,24]. Also, as already published by our group, lifestyle and physical activity may influence the level of hepatic steatosis and the composition of microbiota [24].

The literature reports how diets enriched in orange or specific vegetables lead to improvements in insulin sensitivity and a decrease in visceral fat accumulation [25,26]. The advantages of a higher orange intake in MAFLD patients include improvements in their metabolic homeostasis, mainly due to the presence of polyphenols and, in particular, to their anti-inflammatory effects. These compounds act by inhibiting de novo lipogenesis and stimulating β-oxidation [27].

Regarding micronutrient intakes and anthropometric variables, as evidently supported by our rotating factor and DAPC analyses, three factors emerged as being associated with the entire set of data and gathered variables with highly discriminant statistical weight. In particular, variables in the first factor are responsible for the shifting that marks the orange-treated group in the DAPC plot and account for an increase in SFA, palmitic, stearic, and arachidonic acid values. Significant differences in liver fat content and arachidonic acid-derived lipid mediators are indicative of how the fatty acid variations may lead to liver fat deposition and to increased inflammatory mediators derived from fatty acids.

Our previous study found that daily orange consumption can reduce the prevalence of hepatic steatosis in MAFLD subjects, improve liver function, and control blood glucose levels [28]. Appendix A shows a significant decrease in CAP score, Aspartate Aminotransferase (AST), and Gamma-Glutamyl Transferase (γGT) in the treated group.

Giving a closer look at taxonomic annotations, MAFLD patients at T2 reported increased abundances of *Methanosphaera*, *Eggerthella*, and *Oscillospira*, whereas *Senegalimassilia*, *Negativibacillus*, and *Eubacterium ventriosum* group decreased. Although in significant lower abundance, the presence of the genus *Methanosphaera* has been detected in human stool [29], and its presence seems to activate monocyte-derived dendritic cells after *Methanosphaera stadtmanae* phagocytosis [30]. Its association appears to be controversial, since this taxa has been linked to inflammatory bowel disease (IBD), where it can be detected by the human immune system and might play a role in pathogenic processes in pathologies like IBD [31], whereas it resulted in being linked to a lower likelihood in children suffering from asthma, demonstrating its tolerogenicity in early life [32].

The increase in *Eggerthella* has been linked to the metabolism of bioactive secondary plant compounds, including resveratrol and daidzein from soybeans [33]. It is noteworthy that a gut microbiota consortium containing *Eggerthella* sp. *SDG-2* was recently shown to improve NAFLD [34]. Moreover, after probiotics supplementation in IBS patients from a Chinese cohort, the abundance of *Eggerthella* genus was significantly increased and associated with the improvements in symptom scores [35].

In a recent pediatric study on MAFLD, an important decrease in *Oscillospira* spp., together with *Akkermansia* spp., *Lachnospira* spp., and *Faecalibacterium* spp., was found in comparison with a non-MAFLD group [36]. The increase in our orange-treated group can be thus indicative of a restoration of gut homeostasis. On the other hand, the decreased abundance of *Senegalimassilia* is in line with the recent literature evidence on NAFLD cohort fecal microbiota that reported a negative association of this taxa with the pathology [37]. Similarly, the presence of the *Negativibacillus* genus has been identified as a marker for NAFLD occurrence and progression [38]. Thus, before delving the discrepancies in terms of VOC from fecal and urine samples, it is important to underline how the MAFLD metataxonomics profiles, in common with the metabolomics results, are indicative of an improvement after orange treatment.

In our urine samples, an increase in Camphene concentration was found, which is in line with the recent literature evidence. This terpenoid is one of the major components of their essential oils with antibacterial, antifungal, antioxidant, anti-inflammatory, and hypolipidemic activities [39], and has been found to be the attenuator of hepatic steatosis [40].

2-Propanol, 2-methyl- VOC is most probably metabolized in its oxidized product 2-methyl-1,2-propanediol, which subsequently leads to the formation of 2-hydroxyisobutyrate, normally eliminated with urine [41]. This volatile was investigated in a study delving a set of colorectal cancer patients, and was found to be present both in cancer versus non-cancer profiles [42].

We detected an increase in Phenol, 2-methoxy-4-(1-propenyl)-, also known as isoeugenol, in part of the essential oil compounds. With the aim of improving gut health, this molecule has been tested for its antimicrobial activity against a selection of human pathogens, including *C. difficile*. Together with geraniol, it has been indicated for its potential use in human prophylaxis [43]. Another decreased VOC in T2 orange-treated samples, dodecane, has been associated with *C. difficile* infection, but, as the authors reported, its classification as a potential biomarker needs to be carefully evaluated because of its higher abundance in ambient air than in sample headspace [44].

When the panel of statistically significant fecal VOC was inspected, all (a total of seventeen) but one resulted in a decrease because of the T2-treated group comparison versus T1 samples in the orange-treated arm and T2 controls. Specifically, the comparison between pre- and post-orange-treated samples resulted in four downregulated VOC concentrations (1-Hexadecanol; 5-Hepten-2-one, 6-methyl-; 2,6-Octadienal, 3,7-dimethyl-, (E)-; 1H-Indole, 2,3-dihydro-4-methyl-). The literature evidence has reported that 1-hexadecanol was significantly higher in the saliva of individuals with cirrhosis compared with hepatocellular carcinoma [45]. As we previously reported [46] in a study on a NAFLD patient cohort, the increase in 5-Hepten-2-one, 6-methyl- was linked to a disequilibrium in polar amino acid in the serum liver fibrosis and was linked to a synergistic effect of the diet plus physical activity. The result of an oxidative cleavage of carotenoid has been found to be a major VOC in algae [47], and, in particular, in *Spirulina platensis* [48].

An increase in 2,6-Octadienal, 3,7-dimethyl-, (E)-, also known as citral, was found versus T2 controls, and has been found both in in vitro and in vivo experiments to be the most important discriminative metabolite marker in MAFLD patients exhibiting a higher cardiovascular disease risk [49]. Together with ginger essential oil, citral counteracts high blood TMAO concentration and modulates gut microbiota richness by decreasing cardiovascular-related taxa [50]. Another positive effect observed is the decrease in Ketene after treatment; this molecule is among the aldehyde metabolites induced by small molecule tyrosine kinase inhibitors (TKIs), and, as a highly reactive molecule, can react with lysine and arginine residues [51]. Our analyses revealed, also, a positive correlation between *Senegalimassilia* genus and ketene, which both decreased after treatment.

We detected a decreased ppm concentration of Methyl tetradecanoate (known as myristic acid, methyl ester). SFAs induce apoptosis in hepatocytes, and specifically myristic acid has been proposed as a biomarker of NASH [52]. Moreover, a significative higher consumption of myristic acid was observed in patients affected by fibrosis [53]. Further evidence of the positive impact of orange treatment appears after a linalool increase. This VOC proved to attenuate lipid accumulation and oxidative stress in metabolic dysfunction-associated steatosis liver disease [54], and significantly inhibited lipid production [55].

## 5. Conclusions

Our clinical trial reported significant changes in specific metataxonomics and metabolomics profiles derived from MAFLD patients who followed a dietary regimen enriched in “Navelina” oranges compared with control samples. Stratification analyses justified the divergence of the sub-cohort of MAFLD patients who were administered with oranges.

Specific taxa/VOCs were identified based on pairwise corrected group comparisons. As future perspective we aim at investigating the metatranscriptomics profiles from the same patient cohort.

The multiomics approach here applied is one of the strengths of this clinical trial data inspection. The analysis workflow that we followed led us to have a multifaceted point of view useful to understanding the contribution of a single set of variables. The innovative statistics approach based on a rotating factor analysis and on the computing of an “*a priori*” sample grouping grant the avoidance of any forced belonging of a sample group. The sample size was calibrated to account for the effect size, power, and the correlation among measures on the same subject.

Although the number of samples is acceptable in terms of statistical power, we are planning to increase it by using a bigger patient cohort. Moreover, we aim at filling the knowledge gap between the metabolite and microbiota’s intrinsic functions by inspecting, in the next future, the connection between metatranscriptomics and metabolomics data on an enlarged MAFLD cohort.

## Figures and Tables

**Figure 1 nutrients-16-03543-f001:**
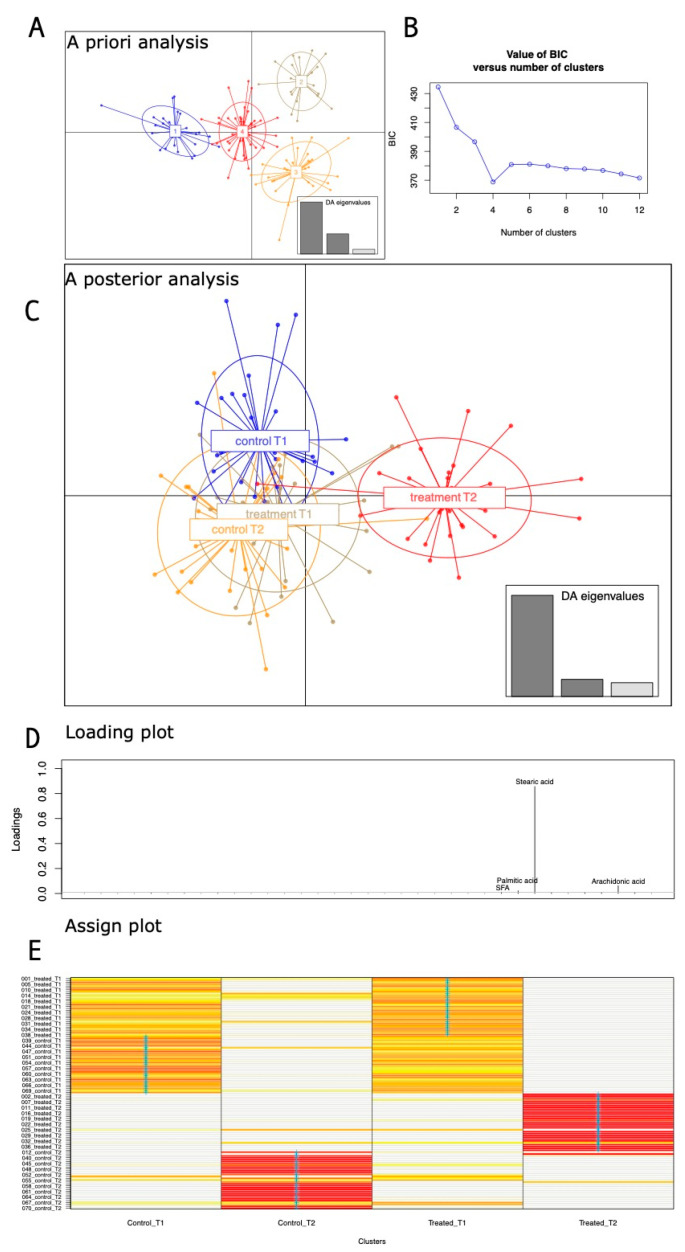
(**A**) The a priori and a posterior DAPCs based on anthropometric and micronutrients/vitamins varying in dietary regimens. (**B**) The minimum in the BIC curve indicated the number of identified clusters in the a priori analysis. (**C**) A posterior DAPC analysis based on known group belonging. (**D**) Loading plot with variables that most contributed to the analysis. (**E**) Assign plot; proportions of successful reassignments: heat colors represent membership probabilities (red = 1, white = 0, orange/yellow = non completely succeeded reassignment) and blue crosses represent the DAPC prior cluster.

**Figure 2 nutrients-16-03543-f002:**
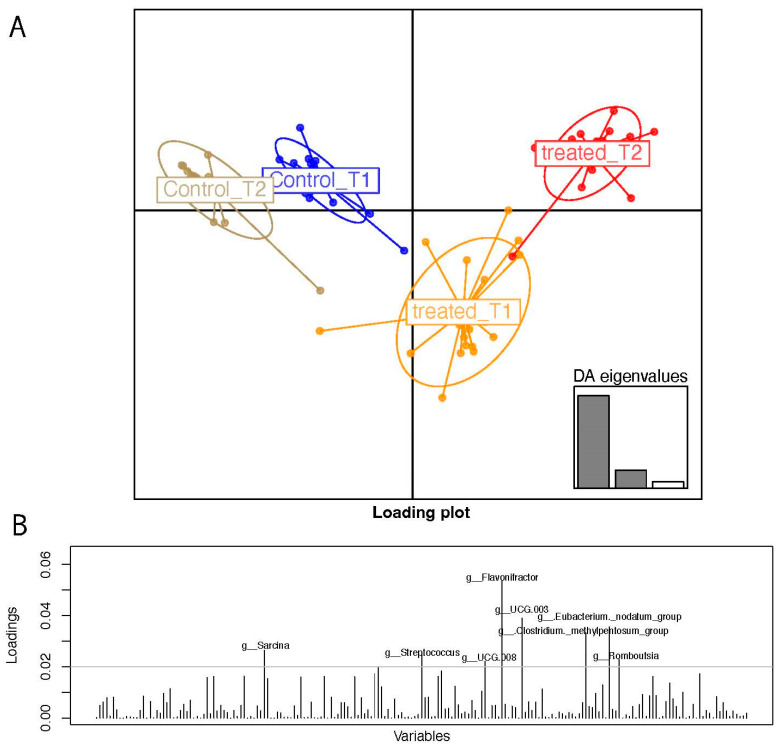
(**A**) DAPC based on genus relative abundances. (**B**): DAPC loading plots reporting most impacting genera over the arbitrary threshold of 0.02.

**Figure 3 nutrients-16-03543-f003:**
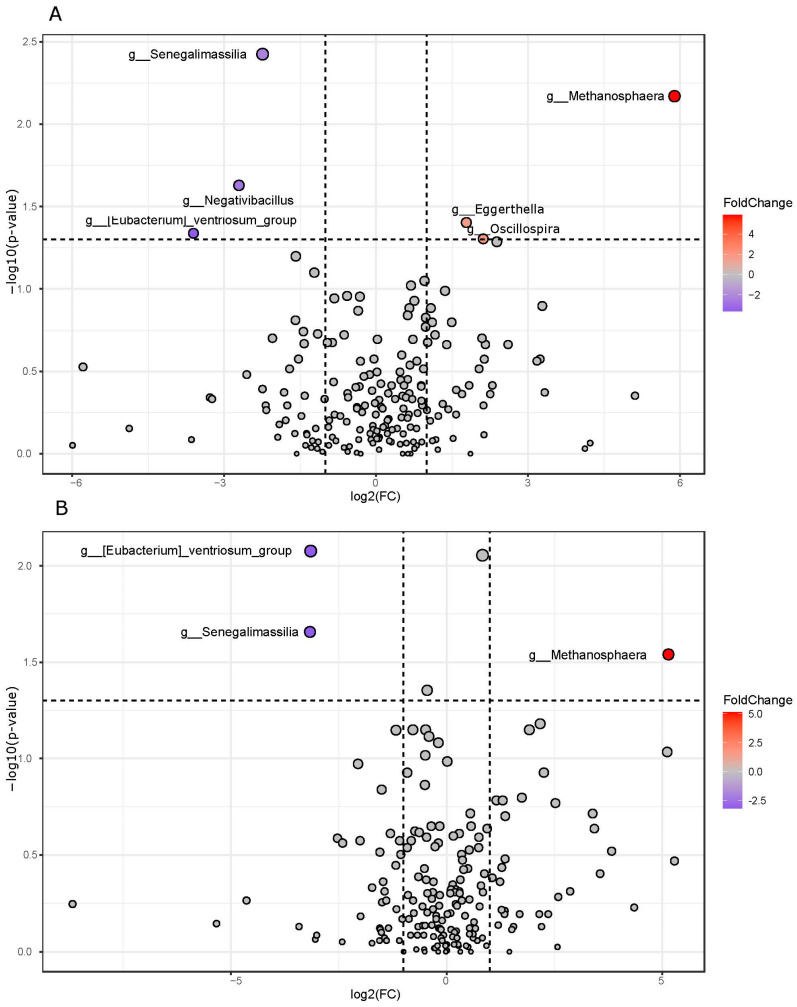
Group comparison based on the Welch test joined with the fold change analyses. Panel (**A**): Pairwise comparison between T2 orange-treated and T2 control. Panel (**B**): Pairwise comparison between T2 orange-treated versus controls. For both the panels, increased (red) and decreased (violet) in orange-administered samples at T2 genera were compared with T2 control (Panel (**A**)) and control samples at T1 (Panel (**B**)), respectively.

**Figure 4 nutrients-16-03543-f004:**
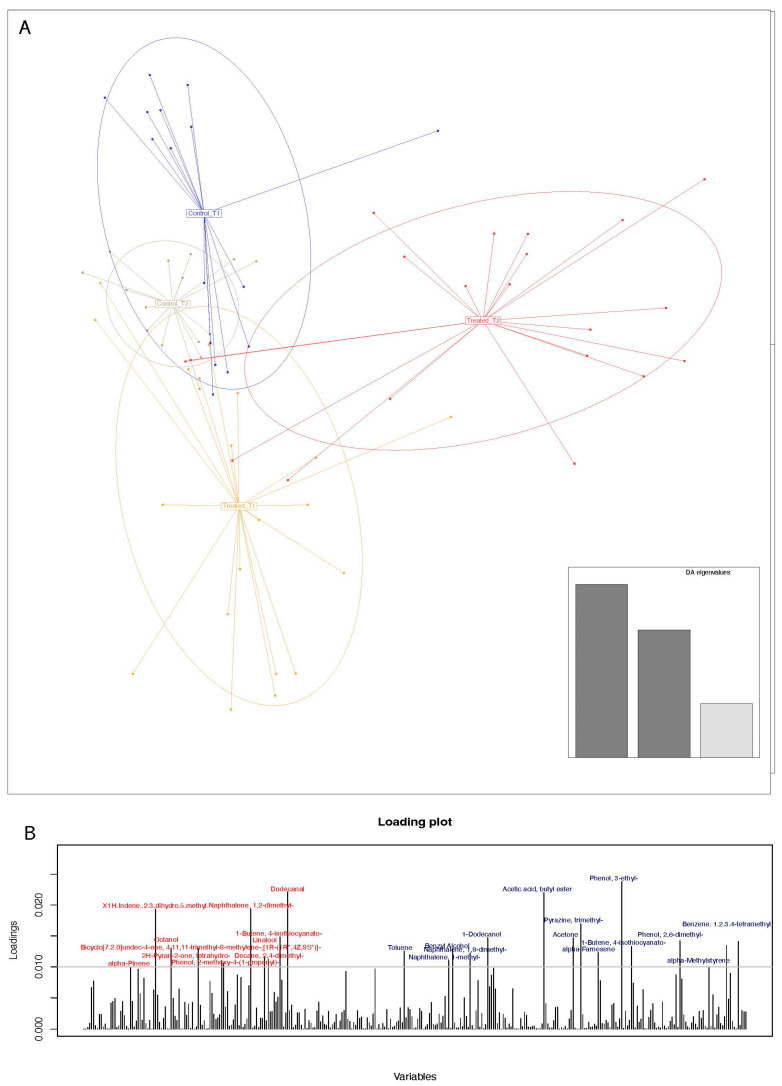
Urine and fecal VOC DAPC plot. Matrices of ppm concentration values from each identified VOC have been merged and used as inputs for the DAPC. (**A**): DAPC plot with used eigenvalues. (**B**): DAPC loading plot with most contributing VOCs above the 0.01 arbitrary threshold. Urine and fecal VOC names have been marked by red and blue fonts, respectively.

**Table 1 nutrients-16-03543-t001:** Rotating factor analysis table. Factor loadings (−1, 1) plus uniqueness (0–1 interval, where 1 is commonality) have been reported for each one of the analyzed anthropometric and nutrient/mineral variables. Variables in bolds satisfied two different criteria: a loading greater than 0.5 and a uniqueness lower than 0.5.

	Factor 1	Factor 2	Factor 3	Uniqueness
Height	−0.064699529	0.06826696	0.178581556	0.959115368
Weight	0.126617531	0.2365219	0.125839197	0.912321273
BMI	0.193744351	0.14582586	−0.024936679	0.940697259
Proteins	0.285081134	0.23399605	0.47936257	0.63412446
Lipids	0.424481027	0.07006638	**0.64060739**	0.404533298
Calorie	0.506103266	0.35677247	0.044906034	0.614558334
Proteins	−0.007605841	0.04698285	0.507746945	0.739936533
Lipids	0.236640486	−0.1452059	**0.807417816**	0.270992434
Carbohydrates	−0.19903003	0.11616393	**−0.970562583**	0.005
Amido	−0.001685977	−0.0974902	−0.530899533	0.708639495
Sugars	0.160694805	0.4711531	−0.292088125	0.666855951
Total fiber	0.070447191	**0.79506517**	−0.18020291	0.330408794
Na	−0.023490989	0.4842104	0.017883766	0.764596798
K	0.113929299	0.64016249	−0.000987283	0.577349561
Iron	0.104782206	0.45273011	0.104974903	0.773027662
Calcium	0.10083975	**0.74665784**	−0.128002353	0.415941726
Phosphorous	0.069290061	**0.77494489**	0.002742983	0.394648949
Thiamine	0.150254483	**0.70648018**	0.163626428	0.451516532
Riboflavin	0.150915412	**0.8769164**	0.230477158	0.155121734
Niacin	0.076939901	**0.6948258**	0.246204482	0.450680114
Vit.A.	0.054915714	0.69734496	0.020818834	0.510275315
Vit.C	0.119471353	0.43628544	0.098776716	0.785623631
Vit.E	0.086494254	0.19081292	0.527066423	0.678332102
Water	0.074253403	0.82138447	−0.022175257	0.319320812
SFA	**0.98497784**	0.141914	0.095471251	0.005
Palmitic acid	**0.984976759**	0.14174306	0.095671108	0.005
Stearic acid	**0.984794254**	0.14417705	0.093389265	0.005
MUFA	**0.984955587**	0.14201112	0.095558933	0.005
Palmitoleic acid	**0.984548983**	0.14297274	0.097210616	0.005
Oleic acid	**0.984958489**	0.14199322	0.095557386	0.005
PUFA	**0.984973149**	0.14202627	0.0953504	0.005
Arachidonic acid	**0.984919328**	0.14188511	0.09528714	0.005
Eicosapentaeic acid	**0.984934595**	0.14200122	0.095758523	0.005
IPAQ	0.190713702	−0.0178699	−0.018727683	0.962848331

**Table 2 nutrients-16-03543-t002:** Untargeted metabolomic pairwise Welch comparison joined with fold change analysis. Untargeted metabolomic significant variables (FDR corrected) have been compared between orange-treated samples (T2) versus T2 controls and T1-treated samples. Following the direction of comparison, increased and decreased Log2 fold change values are relative to the orange T2-treated group.

**Group Comparison**	**URINE VOC**	**FC**	**log2 (FC)**	**P Adjusted**	**−LOG10 (p)**
T2 treated vs. T2 control	Dodecane	0.3338	−1.583	0.036061	1.443
T2 treated vs. T1 treated	Camphene	0.19777	2.3381	0.010314	1.9866
	Phenol, 2-methoxy-4-(1-propenyl)-	3.7734	1.9159	0.022719	1.6436
	2-Propanol, 2-methyl-	0.43842	−1.1896	0.022719	1.6436
**Group Comparison**	**FECES VOC**	**FC**	**log2 (FC)**	**P adjusted**	**−LOG10 (p)**
T2 treated vs. T2 controls	1-Hexadecanol	0.25404	−1.9769	0.062675	1.2029
	5-Hepten-2-one, 6-methyl-	0.33325	1.5853	0.062675	1.2029
	2,6-Octadienal, 3,7-dimethyl-, (E)-	0.334	1.5821	0.062675	1.2029
T2 treated vs. T1 treated	Ethanone, 1-(1-cyclohexen-1-yl)-	0.11429	−3.1292	0.0017474	2.7576
	1H-Indole, 2-methyl-	0.12487	−3.0015	0.0033397	2.4763
	Ketene	0.25691	−1.9606	0.0054678	2.2622
	5-Hepten-2-one, 6-methyl-	0.49425	1.0167	0.0065457	2.184
	2-Propanamine, 2-methyl-	0.33833	−1.5635	0.016406	1.785
	Methyl tetradecanoate	0.28477	−1.8121	0.025727	1.5896
	Acetamide	0.44592	−1.1651	0.030388	1.5173
	Phenol, 2,6-dimethyl-	0.30373	−1.7191	0.031394	1.5032
	Dodecanoic acid, ethyl ester	0.34307	−1.5434	0.033506	1.4749
	Mequinol	0.45427	−1.1384	0.03692	1.4327
	Pyrazine, 2,6-dimethyl-	0.011799	−6.4052	0.039344	1.4051
	Linalool	0.3135	1.6735	0.048158	1.3173
	Propanal, 2-methyl-	0.39026	−1.3575	0.048158	1.3173

## Data Availability

The obtained 16S rRNA fastQ raw sequences are available from the NCBI Bioproject database. The project has a temporary submission PRJNA1163062.

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
