# Peer review of "Metataxonomics and Metabolomics Profiles in Metabolic Dysfunction-Associated Fatty Liver Disease Patients on a “Navelina” Orange-Enriched Diet"

_nutrients, 2024, doi:10.3390/nu16203543_

Round 1

Reviewer 1 Report

Comments and Suggestions for Authors

The manuscript is very interesting because it focuses on how adding a natural food to the diet can produce favorable changes. However, before it is published, the authors must improve some items.

1)Was the statistical analysis performed blindly?

2)The authors describe in methodology that blood samples and anthropometric measurements were taken, however the results are not shown in the manuscript or in the supplementary material.

3)The strengths and weaknesses of the study are missing

4)The results should include some analysis describing whether there is a relationship between the metabolic parameters and the changes found, to evaluate whether the changes found have any clinical importance.

Author Response

Reviewer 1

The manuscript is very interesting because it focuses on how adding a natural food to the diet can produce favorable changes. However, before it is published, the authors must improve some items.

We really thank the reviewer for her/his opinion on our paper.  We used her/his precious  reported corrections/suggestions to improve the whole manuscript.

1)Was the statistical analysis performed blindly?

Many thanks for this comment and for have offered us the possibility to better deep this issue.

Yes, the statistically analyses were blindly performed. The rotating factor statistic uses a blind approach to detect most impacting variables, described in terms of weight and uniqueness. As far as it concern the Discriminant Analysis of Principal Components, the “a priori” clustering was obtained without any sample group belonging information. The resulting assign plots show the fitting between this analysis results (based on the baiyesian information curve – BIC ) and the ones coming from the “a posterior” assignment based on the known groups. Once ascertaining that samples cab be clustered based on these above mentioned specific analyses, we moved on group pairwise comparisons to detect statistically significant changes in terms of variables.

2)The authors describe in methodology that blood samples and anthropometric measurements were taken, however the results are not shown in the manuscript or in the supplementary material.

Many thanks. A supplementary table reporting anthropometric and clinical variables from enrolled patients at baseline and after 28 days of treatment. Please see lines 398-401 and 497-498 for the table caption.

3)The strengths and weaknesses of the study are missing

We are grateful to the reviewer for her/his comment. We added strengths and weaknesses in the conclusion section. Please see lines 484-495.

4)The results should include some analysis describing whether there is a relationship between the metabolic parameters and the changes found, to evaluate whether the changes found have any clinical importance.

Many thanks for highlighting this lack in the followed workflow. We finalized this task by using a Person’s correlation among variables and subsequently by modifying figure and the manuscript main text. Only significant and strong correlations have been reported as results. Please see lines 358-368, lines 466-467 in the discussion section, and 503-509 for the figure caption.

Reviewer 2 Report

Comments and Suggestions for Authors

This is an interesting study with potential value in improving the health status of patients with metabolic syndrome/steatosis, through dietary adjustment. The quality of the manuscript is good, as well as the research design and methods employed. 

However, in the GC/MS method description section, more details should be provided about the MS parameters -i.e., electron energy (70 eV?), mas range detection parameters, SIM or ion scan method,  etc. Also more details should be provided about the chromatogram analysis -what software and if a mass spectral database was used, etc.

Author Response

Reviewer 2

This is an interesting study with potential value in improving the health status of patients with metabolic syndrome/steatosis, through dietary adjustment. The quality of the manuscript is good, as well as the research design and methods employed.

We really thank the reviewer for her/his opinion on our paper.  We have modified the manuscript body text accordingly.

However, in the GC/MS method description section, more details should be provided about the MS parameters -i.e., electron energy (70 eV?), mas range detection parameters, SIM or ion scan method,  etc. Also more details should be provided about the chromatogram analysis -what software and if a mass spectral database was used, etc.

Many thanks. We are sorry for this missing information. We added a more punctual method description relative to the GC/MS used methodology. Please see lines 220-227 in the M&M section.